# Changes in Fitness of Rural Primary School Students from Southwest China after Two-Year’s Nutrition Intervention

**DOI:** 10.3390/nu13103544

**Published:** 2021-10-09

**Authors:** Ran Zhao, Qian Gan, Zhuolun Hu, Peipei Xu, Li Li, Titi Yang, Hui Pan, Xiaoqi Hu, Qian Zhang

**Affiliations:** 1National Institute for Nutrition and Health, Chinese Center for Disease Control and Prevention, Beijing 100050, China; sabrina19840206@163.com (R.Z.); ganqian@ninh.chinacdc.cn (Q.G.); xupp@ninh.chinacdc.cn (P.X.); lili1@ninh.chinacdc.cn (L.L.); yangtt@ninh.chinacdc.cn (T.Y.); panhui@ninh.chinacdc.cn (H.P.); huxiaoqee@163.com (X.H.); 2Beijing Chaoyang District Taiyanggong Community Health Service Center, Beijing 100028, China; 3Department of International Health Human Nutrition, Bloomberg School of Public Health, Johns Hopkins University, Baltimore, MD 21205, USA; zhu27@jh.edu

**Keywords:** height, weight, physical development, physical fitness, nutritional status, China

## Abstract

Children in China’s poor rural areas often have insufficient protein and micronutrient intake. There is little research about the effect of milk and egg supplementation published on these children. A prospective randomized controlled trial was applied to evaluate the effect of milk and egg supplementation on the growth and fitness of poor rural primary school students in southwest China whose physical development was below national averages. A total of 955 healthy students aged 6–13 years old were recruited. The intervention group (538) received 200 g milk and 50 g braised egg at each school day, while the control group (417) kept their normal diet and received no extra supplementation. Serum vitamin D levels were measured by an enzyme-linked immunosorbent assay. A mixed linear model with repeated measures was performed to analyze the efficacy of the supplementation. Statistically significant interactions between groups and time were seen in weight in boys, but not in girls. Significant improvement in vitamin D levels, the broad jump, and the 8 × 50 m shuttle run were observed in both genders. Therefore, the supplementation of egg and milk for two years might have a positive effect on growth and physical fitness and decreasing vitamin D deficiency in poor rural Chinese children.

## 1. Introduction

Childhood is the most critical period for physical and mental development [1]. Unbalanced nutrition in children not only hinders physical growth and increases the risk of disease and death, but also affects their ability to work in adulthood, which would also have a negative impact on economic development. According to The State of the World’s Children 2019, almost 200 million children under the age of five suffered from stunting where height is below the age-gender-specific height range of the screening standard) or wasting (the BMI is below the age-gender-specific BMI range of the screening standards), while at least 340 million suffered from hidden hunger as the result of micronutrient deficiencies [2]. Children living in poor rural areas in China still experienced undernutrition in the past ten years due to the tough natural environment and undeveloped local economy. The Chinese Nutritional Surveillance Report in 2010–2012 indicated wasting (10.3%) and stunting (4.6%) rates of children 6–11 years old in rural areas were much higher than for children of the urban communities (7.5%, 1.3%) based on the Chinese nutritional standard [3]. Children in rural areas had more serious inadequate protein and micronutrient intakes in minerals such as calcium, vitamin A and vitamin D [4,5].

Milk and eggs were good sources of protein and calcium and had been used in some poor rural areas of China with the intention of improving children’s nutrition status. Several studies indicated milk intake was positively associated with body height and weight in children [6,7]. Several publications indicated the supplementation of eggs or animal-source food for children might improve nutritional and health status in low- and middle-income countries (LMICs) [8,9,10]. Several clinical trials have been published on how both milk and egg supplementations would influence physical growth in children. Nevertheless, there have not been enough studies on how these supplementations would affect body fitness [11,12].

Therefore, this study proposes to measure the effect of milk and egg supplementations on poor rural children’s growth and development through a two-year prospective randomized controlled trial in Tianyang County, Guangxi Zhuang Autonomous Region, a poor rural county in southwest China.

## 2. Materials and Methods

### 2.1. Subjects and Study Design

This two-year intervention trial was conducted from 2013 to 2015 with data collection in April 2013 for the baseline, April 2014 for the first year, and April 2015 for the second year. Referring to the cross-sectional study on growth and development of Chinese Han students [13], we used the formula N = 2(Zα + Zβ)^2^δ^2^/d^2^ (α = 0.05, β = 0.20, δ = 7, d = 1.5) and calculated the sample size in this subject. We considered the loss of follow-up in the investigation to increase the sample size by 20%. The calculation was 409 in both intervention group and control group. The total sample size was not less than 818.

For the intervention group, four primary schools in the rural township of Tianyang County were randomly selected. For the control group, four other primary schools were randomly selected in the same township in order to ensure a similar socioeconomic level, teaching quality, and scale. In each primary school, one class with about 30 healthy children aged 6–11 years old was randomly selected from first grade to fifth grade. Students were excluded if they were suffering from diseases that may interfere with normal growth and development. In total, 955 children (538 in the intervention group and 417 in the control group) were involved in the study (Figure 1). This county was one of the most underdeveloped counties in China with a GDP of 23,970.78 CHY (3835.32 USD) per person in 2013, while the national average was 41,907.59 CHY (6705.21 USD) [14].

From April 2013 to April 2015, children usually had lunch at school at 11:30 am. At 2:30 pm on each school day, the schoolmaster/teacher responsible distributed the milk and eggs to students in the intervention group, who consumed it on the spot. The milk was ultra-high temperature (UHT) school milk with different flavors (net weight 200 g), and the eggs were pre-packaged and braised (net weight 50 g). After correcting for weekends, public holidays, winter and summer holidays, students in the intervention group were provided with milk and eggs for 380 school days over a two-year period. In the control group, the children maintained their normal diet without any intervention measures. Table 1 provides a comparison between the milk and egg nutrient composition in this supplementation trial and the Recommended Nutrient Intake (RNI) for Chinese children aged nine years old. The study protocol was approved by the Ethics Committee of the National Institute for Nutrition and Food Safety, the Chinese Center for Disease Control and Prevention, and informed consent forms were signed by the children’s parents before the study.

### 2.2. Anthropometric Measurement

Trained investigators, using pre-calibrated instruments, measured the fasting weight and height in the morning following the standard procedures. Children were in their underwear and had taken off their shoes and socks. Height was corrected to 0.1 cm, weight to 0.1 kg, and the Body Mass Index (BMI) = weight (kg)/height (m^2^) was calculated. Height and weight were measured at baseline, the first year, and the second year.

### 2.3. Vitamin D

Students in the second and third grades (aged 7–8) each had 5 mL of fasting venous blood drawn in the morning, which was centrifuged to separate the serum. Serum 25(OH)D was measured by an enzyme-linked immunosorbent assay in μg/L at baseline and at the second year. According to the literature, 25(OH)D < 10 ng/mL was used for vitamin D deficiency, 10 ng/mL ≤ 25(OH)D < 20 ng/mL were used for subclinical deficiency, and 25(OH)D ≥ 20 ng/mL for normal vitamin D levels [15].

### 2.4. Physical Fitness

Trained physical education teachers measured the scores on the board jump and 8 × 50 m shuttle run among students in grades two to four at baseline, and at the first and second year.

The broad jump (reflecting lower body explosiveness) was determined as the distance from the trailing edge of the jumper to the trailing edge of the nearest location by a Changcheng Brand 50 m leather tape. The measurement was corrected to 0.1 cm.

A uniformly calibrated stopwatch was used to measure the time of a round trip (reflecting endurance). On the 50 m runway, a 1.2 m high pole was set at 0.5 m and 49.5 m before the (final) point line. The subjects need to run four laps. The measurement result was corrected to 0.1 s. 

### 2.5. 24-h Dietary Recall Method for Three Consecutive Days

At baseline and second year, students in grades three and four had dietary intake surveys taken through questionnaires (24-h dietary recall) which were completed on Sundays, Mondays, and Tuesdays. Before filling out the questionnaire, the investigator explained the method in detail and provided common food samples or pictures. Students were asked to record intake of the type and the quantity of all food including beverages and snacks (except condiments). According to the Chinese Food Composition Table (2002 edition, 2004 edition, and 2009 edition), trained coders coded the questionnaire and calculated the intake of energy and various nutrients.

### 2.6. Statistical Analysis

Descriptive statistics were reported as mean ± SDs, or N (%) unless otherwise indicated. Quantitative data on this subject had been tested for normality before analysis. A *t*-test was used for normal distribution and a rank sum test was used for non-normal distribution. A Chi-square test was used to compare the differences of qualitative data.

Since subjects repeated the same measures for three consecutive years, a mixed linear model with repeated measures was adapted to analyze the efficacy of the intervention. Subjects in school were defined as random effects. Time was defined as level one unit and tested within-individual differences. Group was defined as a level-two unit and tested for between-individual differences. The interaction effect between group and time was tested by the changes over time in the difference between the two groups. Cohen’s f^2^ was used to measure fixed effects and ICC was used to measure random effects. A *p*-value smaller than 0.05 would indicate an effective intervention. All data were numbered and entered into EpiData. Statistical analysis was performed with SAS (SAS 9.4 for Windows, SAS Institute, Inc., Cary, NC, USA). 

## 3. Results

### 3.1. The Characteristics of the Participants

Out of the 955 children enrolled at baseline, 769 children were followed up in the two-year intervention study (Figure 1). Reasons for loss to follow-up were due to graduation of senior students (75.3%) and transfer of school (24.7%). Compared to children who stayed for the entire study, lost to follow-up children had higher age, height, weight, and BMI at baseline with higher grades (*p* < 0.05) (Table 2).

Dietary protein, fat, and iron were higher in the intervention group than in the control group at baseline (*p* < 0.05) (Table 3). All nutrient intake, including the daily supplemented egg and milk, was higher in the intervention group than those in the control group in the second year. The students’ physical activity time was significantly higher than the control group at baseline but was no different in the second year.

### 3.2. Effect of Intervention on Height and Weight

For height in boys and girls, no significant difference was observed between the intervention group and the control group at baseline. The effect of time at the first year and the second year was statistically significant for both genders (*p* < 0.001). No significant effect was observed for groups or interaction between time and group. Minimum effect of intervention on height (M:3.57 × 10^−16^, F:3.41 × 10^−16^) was observed.

The data of weight had a large degree of intra-individual correlation (ICC: M for 0.69, F for 0.66), but the effect of intervention behavior on body weight was small (f^2^: M for 0.07, F for 0.05). For weight at baseline, no significant differences were observed between the intervention group and control group either in boys or in girls. The time effect was significant after the first and the second year for the weight of boys and girls (all *p* < 0.001). The interaction between time and group was significant after the first and second year for boys’ weight (*p* < 0.001). Boys in the intervention group gained 0.6 kg (*t* = 3.80, *p* < 0.001) more weight after the 1st year on average, and 0.7 kg more after the second year than that of the control group (*t* = 1.99, *p* = 0.046). While no statistical significance was observed for a group effect on the weight of either the boys or the girls, neither was the interaction between time and group on girls’ weight. (Table 4, Table A1 and Table A3). 

### 3.3. Effect of Vitamin D

At baseline, there was a statistically significant difference in the proportion of vitamin D deficiency between the intervention group and the control group, with the intervention group being higher (*p* = 0.007). Subclinical deficiency (17.1%) and deficiency (44.6%) in the intervention group were higher than those in the control (11.2%, 38.8%). There was no significant difference in the proportion of vitamin D deficiency between the two groups in the 2nd year (*p* = 0.542) (Table 5). 

### 3.4. Effect of Intervention on Physical Fitness

For boys’ and girls’ broad jump scores, no significant differences were observed between the intervention group and the control group at baseline. The effect of time intervention after one year and two years was significantly increased in the boys’ intervention group (*p* < 0.001) as well as the girls in the second year (*p* = 0.01). The boys’ and girls’ scores in the intervention group increased 11.7 cm more (*t* = 4.45, *p* < 0.0001) and 13.7 cm more (*t* = 4.24, *p* < 0.0001) than those in the control group in the 2nd year, respectively, while the interaction between time and group was significant. From the result of effect size, the intervention had a medium effect on girls’ broad jump (0.19), but little effect on boys’ broad jump (0.01).

For boys’ and girls’ 8 × 50 m shuttle run scores at baseline, no significant difference was observed between the groups. Boys’ scores in the intervention and control groups increased over the duration of the study (*p* < 0.001). The trend of this index increase between the two groups was statistically significant. The intervention group of boys’ and girls’ scores had a 12.8 s increase (*t* = −5.11, *p* < 0.0001) and a 23.8 s increase (*t* = −7.39, *p* < 0.0001), respectively, compared with the control group in the first year, and had an 18.4 s (*t* = −7.65, *p* < 0.0001) and a 28.4 s increase (*t* = −10.14, *p* < 0.0001), respectively, in the second year (Table 6, Table A2 and Table A3). 

## 4. Discussion

This intervention study demonstrated that supplementation of 200 g milk and 50 g egg at each school day for two years promoted the increase of weight in boys, but not in girls. It decreased vitamin D deficiency and improved physical fitness with regard to a longer broad jump and a faster 8 × 50 m shuttle run to some extent.

Children in China’s poor regions have been in lower weight and height brackets compared to their urban counterparts [15]. Poor rural students were generally in “nutritional poverty” [16,17]. Their height and weight were below national averages for children living in rural areas during the study period. Compared to the Monitoring of Nutrition and Health Status of Chinese Residents (2010–2013 Synthesis Report) with averages for rural students of the same gender and age, baseline data indicated a 6–8 cm gap in height and a 4–7 kg gap in weight [3]. Therefore, it was vital to improve the nutritional status of children in underdeveloped rural areas. In the Chinese Dietary Guidelines for school-Age Children, the recommended daily intake was 300 g of milk and 25–50 g of eggs [18]. Children in rural areas generally have a low intake of dairy products. In this study, providing 250 g of milk and one egg to rural children satisfied 83% of the milk and egg intake requirements. This combination provides 53.3% of energy and 29.7% of protein required, and it increases the average weight of the sample. After two years of milk and egg supplementation, the gap in height and weight between subjects and national rural children still existed but was narrowed (2–3 cm, 1–2 kg).

Our research has observed effects on weight but not on height. Similar results were observed in a six-month supplementary study among 241 Ugandan students (age 6–9) [19]. In their study, the groups of one-egg supplementation and two-egg supplementation had a greater increase in children’s weight and mid-upper arm circumference compared to the zero-egg group, but not in height. Some literature has suggested that development in height should be considered over the long run, for instance at least for the last three years among school-age children [20]. The intervention involved in this study included school days only, thus shortening the duration of the intervention. This could contribute to not finding significant effects on height.

We also observed students’ subclinical vitamin D deficiency, and it was significantly decreased compared to the baseline. A survey conducted on students aged 9–12 in Madrid also indicated a significant difference in vitamin D intake and serum levels between the groups with ≥0.5 eggs intake per day < 0.5 egg intake per day. The former had a lower risk of vitamin D deficiency (OR = 0.41, 95% CI: 0.19, 0.88). Egg yolk is considered one of the most important sources of vitamin D in the diet, and it contains a high amount of 25-hydroxyvitamin D [21]. It proved that the consumption eggs every day can prevent vitamin D deficiency, which was consistent with the results of this study [21].

Several pieces of the literature indicated that protein supplementation can increase the muscle volume and muscle fiber cross-sectional area [22]. It can also increase muscle strength and muscle explosiveness [23]. Other results in our study published also demonstrated that the combined intervention of milk and egg would facilitate lean body mass growth of boys in rural China [24]. Moreover, past research has found higher vitamin D intake could improve muscle strength [25]. In this study, the broad jump scores reflected the students’ explosive power of the lower limbs. The 8 × 50 m round trips reflected the comprehensive ability of speed, endurance, and coordination. In the final survey, compared with the same age and gender student data of that reported on in the physical fitness and health surveillance of Chinese school students in 2014, the broad jump scores were 2–8 cm higher and the 8 × 50 m round trips performance was 3–8 s faster than the national average [26]. The scores of the intervention group were higher than those of the control group. It may be related to the intake of milk and eggs, which were rich in high-quality protein or able to improve vitamin D status.

The participants from this study only consumed one egg per serving, and this had not exceeded the recommended dietary intake in China. It was indicated that the effect of cholesterol in children might be different from that of adults: excess cholesterol increases the risk of developing cardiovascular diseases and mortality rate in adults but improves the growth of children [27]. Long-term effects on cholesterol levels need to be observed in Chinese children.

There are several strengths and weaknesses in our study. This was one of the first few studies that conducted randomized controlled trials on poor rural children in southwest China with regard to body fitness. Nevertheless, the supplementation only lasted for two years, which cannot demonstrate the long-term influence on participants’ height and cholesterol level. Furthermore, there were differences in the nutrient levels among participants in the control group where they kept their normal diets. The discrepancy in the results of dietary survey data could be related to coding quality control. Future studies can provide the supplementation for a longer period of time to observe the long-term influence on participants’ biometrics and physical fitness. Additionally, researchers can measure the physiological mechanisms of the participants.

Moreover, this intervention might be popularized as an effective method to improve poverty-stricken rural children’s physical conditions. The measures will endorse efforts to further improve children’s nutritional and physical health and achieve nutritional targets. As ordinary foods, eggs and milk can serve as food implementations. The influence and the cost-effectiveness can be observed when they are promoted by policies.

## 5. Conclusions

After the two-year intervention was implemented, the supplementation with eggs and milk had a positive effect on students’ weight and physical fitness. Intervention measures had reduced the vitamin D deficiency rate and improved the students’ physical condition.

The State Council of China introduced the National Nutrition Program (2017–2030) in July 2017. One of the major actions was the student nutrition improvement program. It included guiding students to have a nutritionally beneficial diet. In this study, milk and egg supplementation were likely to help increase children’s weight and improve their physical fitness while reducing undernutrition and vitamin D deficiency. This intervention might be popularized as an effective method to improve poverty-stricken rural children’s physical conditions. The measures will endorse efforts to further improve children’s nutritional and physical health and achieve nutritional targets.

## Figures and Tables

**Figure 1 nutrients-13-03544-f001:**
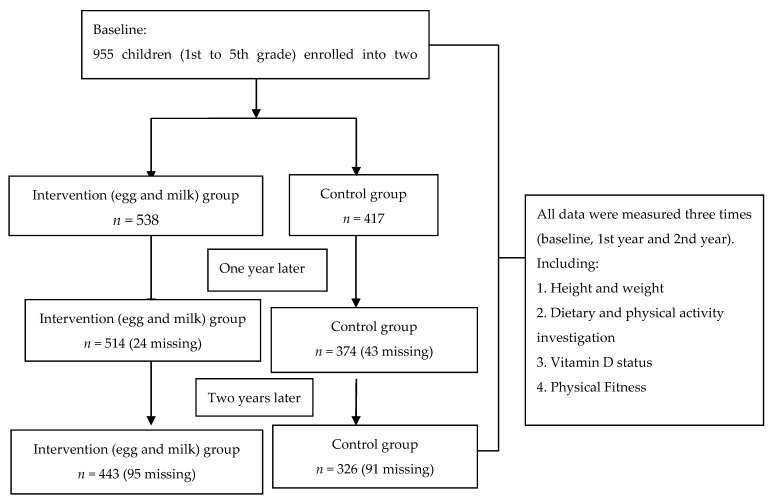
Flow diagram of participation.

**Table 1 nutrients-13-03544-t001:** Comparison of nutrient content of milk and egg with Chinese RNI for nine year old children in the supplementation trial.

	Nutrient Content ^1^			Chinese RNI ^2^ for 9-Year Children		Proportion of RNI (%)
	Milk	Egg	Sub-Total	Boy	Girl	
Energy, kj	578	301	879	1750	1550	53.3
Fat, g	8	5.8	13.8	20–30		55.2
Protein, g	7	6.35	13.35	45 13 1000 500 1000		29.7
Fe, mg	0.2	3.5	3.7			28.5
Ca, mg	213	28	241			24.1
VitA, ugRAE	280	117	397			79.4
Vit B_2_, ug	260	135	395			39.5

^1^ Milk weighed 200 mL, and egg weighed 50 g; ^2^ RNI: Recommended Nutrient Intake.

**Table 2 nutrients-13-03544-t002:** Comparison of characteristics of subjects with completed two years and lost to follow-up at baseline (means ± SDs).

	Completed 3-Year Study	Lost to Follow Up
N	769	186
Boy proportion, % ^3^	50.8	51.6
Age, yr ^1^	8.8 (8.0–9.9)	10.5 (8.5–11.6) *
Height, cm ^2^	125.4 ± 8.0	132.2 ± 11.4 *
Weight, kg ^2^	23.8 ± 4.7	27.5 ± 7.5 *
BMI, kg/m^2 2^	15.0 ± 1.6	15.4 ± 2.0 *

^1^ Rank sum test was used to compare the characteristics of age; ^2^
*t*-test was used to compare the characteristics of height, weight and BMI between the two groups; ^3^ Chi-square test was used for boy proportion; * *p <* 0.05. Notes: Values were means ± SDs or medium (Q1–Q3) as listed. A significant difference was seen in age, height, weight and BMI except boy proportion.

**Table 3 nutrients-13-03544-t003:** Comparison of dietary intake and physical activity between the intervention group and the control group at baseline and second year (mean ± SDs).

	Baseline		1st Year		2nd Year	
	Intervention	Control	Intervention	Control	Intervention	Control
Age, y ^2^	8.9 (8.0–10.1)	9.2 (8.0–10.5) *	9.9 (9.0–11.1)	10.2 (9.0–11.5) *	10.9 (10.0–12.1)	11.2 (10.0–12.5) *
Energy, kcal/d ^1,2^	2033.1 ± 41.1	1943.6 ± 47.9	–	–	1785.6 (1349.8–2212.7)	1582.1 (1203.6–1900.5) *
Protein, g/d ^2^	80.3 (67.4–98.8)	74.0 (63.0–84.9)	–	–	64.6 (46.0–92.0)	52.9 (39.7–62.7) *
Fat, g/d ^2^	39.4 (28.5–45.9)	30.5 (24.7–40.4) *	–	–	55.7 (36.5–77.9)	38.4 (25.5–53.4) *
Fe, g/d ^2^	20.5 (16.4–24.9)	18.4 (15.8–21.6) *	–	–	17.3 (13.2–23.7)	14.5 (10.7–17.9) *
Ca, g/d ^2^	439.6 (278.8–648.7)	414.2 (284.3–541.2)	–	–	320.2 (233.6–438.4)	226.4 (150.6–321.4) *
VitB2, g/d ^2^	0.6 (0.5–0.7)	0.7 (0.5–0.8)	–	–	0.7 (0.5–0.9)	0.5 (0.4–0.7) *
Physical activity time, min/d ^2^	40 (40–40)	40 (40–40) *	–	–	40 (40–60)	40 (30–60)
Secondary time, hr/d ^2^	2 (2–3)	2 (2–3)	--	--	2 (2–4)	3 (2–4)

^1^*t*-test was used to compare the characteristics of energy; ^2^ Rank sum test was used to compare the characteristics of age, protein, fat, Fe, Ca, VitB_2_; * *p* < 0.05. Notes: Values were means ± SDs or medium (Q1~Q3) as listed.

**Table 4 nutrients-13-03544-t004:** Comparison on height and weight between intervention group and control.

	Baseline		1st Year		2nd Year	
	Intervention	Control	Intervention	Control	Intervention	Control
Boy						
Height (cm) ^@, @@^	127.1 ± 8.4	126.8 ± 9.8	131.8 ± 8.7	131.7 ± 10.4	136.5 ± 8.6	135.6 ± 8.5
Weight (kg) ^@, @@^	24.9 ± 5.5	24.2 ± 5.7	28.1 ± 5.9 *	27.2 ± 6.9	31.0 ± 7.2	29.6 ± 6.2
Girl						
Height (cm) ^@, @@^	126.1 ± 9.5	127.2 ± 9.3	132.2 ± 9.1	132.9 ± 9.7	136.9 ± 9.1	138.5 ± 9.0
Weight (kg) ^@, @@^	24.2 ± 5.6	24.7 ± 5.5	28.1 ± 6.9	28.0 ± 6.6	31.1 ± 7.4	32.3 ± 7.5

^*^ Time × group effects: *p* < 0.05. Notes: Values were mean ± SDs. Mixed linear model was used to compare the repeated data; Time effect: ^@^ one year *p <* 0.05; ^@@^ two years *p <* 0.05.

**Table 5 nutrients-13-03544-t005:** Vitamin D deficiency between intervention group and control group at baseline and 2nd year (%)^1^.

	Baseline		1st Year		2nd Year	
	Intervention	Control	Intervention	Control	Intervention	Control
Serum 25(OH)D (μg/L)	15.9 (11.78–27.41)	19.82 (14.99–27.02) *	–	–	23.50 (19.20–28.15)	23.90 (18.80–28.60)
Nutrition Status			–	–		
Normal	38.3	50.0			71.3	69.3
Subclinical deficiency	44.6	38.8	–	–	27.8	28.5
Deficiency	17.1	11.2	–	–	0.9	2.2

Notes: Chi-square test was used to compare the rate of malnutrition between intervention group and control group at baseline and 2nd year; Rank sum test was used to compare the characteristics of Serum 25(OH)D; Values were medium (Q_1_~Q_3_); * *p* < 0.05.

**Table 6 nutrients-13-03544-t006:** Comparison of broad jump and 8 × 50 m round trip between intervention group at baseline, 1st year and 2nd year (means ± SDs).

	Baseline		1st Year		2nd Year	
	Intervention	Control	Intervention	Control	Intervention	Control
Boy						
Broad jump (cm) ^@, @@^	144.7 ± 21.5	154.7 ± 17.8	157.8 ± 22.7	160.5 ± 19.5	171.6 ± 19.2 *	165.0 ± 19.6
8 × 50 m round trips (s) ^@, @@^	134.4 ± 31.6	108.3 ± 13.6	112.3 ± 31.8 *	98.9 ± 15.5	108.9 ± 28.3 *	101.0 ± 14.6
Girl						
Broad jump (cm) ^@, @@^	133.9 ± 23.1	149.0 ± 14.9	144.3 ± 18.3	151.4 ± 18.7	157.7 ± 18.9 *	154.1 ± 15.5
8 × 50 m round trip (s) ^@, @@^	142.4 ± 30.8	112.4 ± 13.9	118.6 ± 35.5 *	107.3 ± 17.5	114.6 ± 30.7 *	107.4 ± 14.4

Notes: Values were means ± SDs. Mixed linear model was used to compare the repeated data; Time effect: ^@^ one year *p <* 0.05; ^@@^ two years *p <* 0.05; Time × group effects: * *p* < 0.05.

## Data Availability

Data available on request due to privacy restrictions. The data presented in this study are available on request from the corresponding author. The data are not publicly available due to privacy.

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
