# Peer review of "Changes in Fitness of Rural Primary School Students from Southwest China after Two-Year’s Nutrition Intervention"

_nutrients, 2021, doi:10.3390/nu13103544_

Round 1

Reviewer 1 Report

The manuscript is of very high quality, the topic of the publication is scientifically and clinically beneficial. I recommend its acceptance for publication after the elimination of formal shortcomings and revision of statistical data processing.

The manuscript presents the results of a nutritional intervention study of Chinese school children (age 6-13), which took place between years 2013 and 2015. The concept of the study (randomized design) and the quantity of observations in the longitudinal follow-up are relevant. The results are valuable and beneficial, also demonstrate the importance of the nutritional influence on various aspects of children's growth and development.

Introduction:

Without comments.

Material and Methods:

Section 2.1

Line 63, how is “normal growth and development” defined in relation to the national reference data?

Section 2.4

Would it be possible to expect differences in normal physical activity between the two groups of children?

Section 2.6

Statistical data processing is not entirely correct.

1) The requirements for the t-test (and partly also for the mixed linear model) is a normal (parametric) distribution of values. Has this condition been tested, or has the data been transformed so that this requirement is met? Not specified.

2) There has not been evaluated effect-size, which is crucial for evaluation of clinical relevance. I consider this to be absolutely essential.

Results:

In the text and tables (including the Appendix) the p-values ​​are not written uniformly, unify.

Table 2. - numebered references in the table legend are not applied in the table itself. It is then illogical to number them at all. In the table, link no. 2 is assigned to "Boy proportion, %", however, in the legend, link no. 3 corresponds to this part? Revise.

Table 4 and 6. - again, the numbered references in the legend are not assigned to any values. Non-standard labelling of statistical significance is used. I recommend using only the asterisk convention.

Table 5. - p-values ​​are missing and inconsistently written (P, p).

Discussion:

The discussion is clear and comprehensive, commenting on all monitored aspects of nutritional intervention.

Conclusion:

Without comments.

Author Response

Comment 1: Section 2.1, Line 63, how is “normal growth and development” defined in relation to the national reference data?

Response: This paragraph explains the entry criteria. If the child suffers from diseases that affect their growth and development, they will not be selected in this project. It is more appropriate to use “diseases that affect growth and development” here and we have revised it.

Comment 2: Section 2.4, Would it be possible to expect differences in normal physical activity between the two groups of children?

Response: We considered this issue. In the children’s questionnaire, we counted physical activity time as a reference application. However, questionnaire data still had limitations. In future research, we will use more actual measured data as a reference.

Comment 3: Section 2.6, The requirements for the t-test (and partly also for the mixed linear model) is a normal (parametric) distribution of values. Has this condition been tested, or has the data been transformed so that this requirement is met? Not specified.

Response: Thank you for your guidance on the article data. We re-checked and verified all the data involved in the article. We ensured that data and analysis methods meet statistical requirements. All the Quantitative data in this subject had been tested for normality. T-test was used for normal distribution and rank sum test was used for non-normal distribution. The Chi-square test was used to compare the differences of qualitative data. We added the detailed description in the “Statistical Analysis” part.

Comment 4: Section 2.6, There has not been evaluated effect-size, which is crucial for evaluation of clinical relevance. I consider this to be absolutely essential.

Response: To make it clearer, we have used the formula N=2(Zα+Zβ)2δ2/d2 (α=0.05,β=0.20,δ=7,d=1.5) and calculated the sample size in this subject referring to the cross-sectional study on growth and development of Chinese Han Students. We have considered the loss of follow-up in the investigation to increase the sample size by 20%. The calculation is 409 in both intervention group and control group. The total sample size is no less than 818.

Comment 5: Results, In the text and tables (including the Appendix) the p-values ​​are not written uniformly, unify.

Response: We unified to the “p-value” format.

Comment 6: Table 2. - numbered references in the table legend are not applied in the table itself. It is then illogical to number them at all. In the table, link no. 2 is assigned to "Boy proportion, %", however, in the legend, link no. 3 corresponds to this part? Revise.

Response: We have revised the number according to the table legend.

Comment 7: Table 4 and 6. - again, the numbered references in the legend are not assigned to any values. Non-standard labelling of statistical significance is used. I recommend using only the asterisk convention.

Response: We used @ and * in order to distinguish time effect and time*group effect. We unified p value and used p <0.05 to indicate statistical significance.

Comment 8: Table 5. - p-values are missing and inconsistently written (P, p).

Response: We unified to the “p-value” format and adjusted table 5 format.

Reviewer 2 Report

You have done good work

Author Response

Thank you for your review. 

Reviewer 3 Report

Introduction

  • Clear purpose/objective
  • Accurate and precise title
  • Not sure if the article is to be read by a layperson, could potentially define “wasting” and “stunting” (Line 33 & 34)
  • “researches” should be corrected to “research” (line 47)
  • Could possibly reword sentence (line 47) to “ Several pieces of research have been published on clinical trials with both milk and egg supplementations in children, however, there has not been enough studies on body fitness”.
  • Sequence of information flows well

Methods

  • Methods are valid, reliable, and repeatable.
  • Clarity is needed on whether control group was age matched and age comparisons should be subsequently be noted in results. (age is an obvious factor in terms of growth and development)
  • Strong sample size of 955 participants total (538 intervention)
  • Good use of figure to show participant adherence/dropout
  • Good flow and use of headings

Results

  • Would be helpful to have figure to show results of tables, to illustrate change over time.
  • Results reflect what was intended to be researched as mentioned in the introduction/methods

Discussion

  • Good interpretation of results
  • Good comparison/contrast to other research
  • Highlights strengths/weaknesses of the study 
  • Does not directly give suggestions for future research

Author Response

Comment 1: Not sure if the article is to be read by a layperson, could potentially define “wasting” and “stunting” (Line 33 & 34)

Response: Stunting is defined as the height below the age-specific height range of the screening standard. Wasting is defined as the BMI below the age-specific BMI range of the screening standards. We added these definitions to Line 33.

Comment 2: “researches” should be corrected to “research” (line 47)

Response: We accepted and corrected it.

Comment 3: Could possibly reword sentence (line 47) to “Several pieces of research have been published on clinical trials with both milk and egg supplementations in children, however, there has not been enough studies on body fitness”.

Response: We accepted and corrected it.

Comment 4: Clarity is needed on whether control group was age matched and age comparisons should be subsequently be noted in results. (Age is an obvious factor in terms of growth and development)

Response: We have added the age information according to the comments.

Comment 5: Does not directly give suggestions for future research

Response: We have added suggestions for future research.

Round 2

Reviewer 1 Report

There are still major shortcomings in the Statistical Data Analysis section. The authors have not completed the revision and validation of the data they write about in the comments.The authors confuse sample size and effect-size. Effect-size is an important indicator of the clinical relevance of data.

Author Response

Thank you for your guidance on the article data. We increased the effect size of the model, cohen's f2 was used to measure fixed effects and ICC (intra-class correlation coefficient) was used to measure random effects. f2≥0.02, f2≥0.15, f2≥0.35 represents small, medium, and large effect sizes, respectively. ICC<0.5, 0.5≤ICC<0.75, 0.75≤ICC<0.9, ICC≥0.9 represents poor, moderate, good, excellent effect sizes, respectively.